# Major Adverse Cardiac and Cerebrovascular Events in Geriatric Patients with Obstructive Sleep Apnea: An Inpatient Sample Analysis

**DOI:** 10.3390/medsci11040069

**Published:** 2023-10-30

**Authors:** Rupak Desai, Sai Priyanka Mellacheruvu, Sai Anusha Akella, Adil Sarvar Mohammed, Pakhal Saketha, Abdul Aziz Mohammed, Mushfequa Hussain, Aamani Bavanasi, Jyotsna Gummadi, Praveena Sunkara

**Affiliations:** 1Independent Researcher, Atlanta, GA 30033, USA; drrupakdesai@gmail.com; 2Department of Public Health, University of Massachusetts, Lowell, MA 01854, USA; saimellach@gmail.com; 3Department of Internal Medicine, Kakatiya Medical College, Warangal 506007, India; saianusha.akella@gmail.com; 4Department of Internal Medicine, Central Michigan University, Saginaw, MI 48602, USA; 5Department of Internal Medicine, Bhaskar Medical College, Hyderabad 500075, India; saketha.pakhal@gmail.com; 6Department of Internal Medicine, Kamineni Institute of Medical Sciences, Narketpally 508254, India; m.aziz17725@gmail.com (A.A.M.); hussainmushfequa@gmail.com (M.H.); 7Department of Internal Medicine, Captain James A Lovell FHCC/Rosalind Franklin University of Medicine and Science, North Chicago, IL 60064, USA; aamani.bavanasi@gmail.com; 8Department of Medicine, MedStar Franklin Square Medical Center, Baltimore, MD 21237, USA; jyotsna.gummadi@medstar.net; 9Department of Internal Medicine, Medstar Medical Group, Charlotte Hall, MD 20622, USA; praveenasunkara@gmail.com

**Keywords:** obstructive sleep apnea, major adverse cardiac events, prevalence, stroke, mortality, predictors

## Abstract

Background: Obstructive sleep apnea (OSA) is associated with an increased risk of major cardiac and cerebrovascular events (MACCE). However, data on the burden and predictors of MACCE in geriatric patients with OSA (G-OSA) remain limited. Methods: Using the National Inpatient Sample from 2018, we identified G-OSA admissions (age ≥ 65 years) and divided them into non-MACCE vs. MACCE (all-cause mortality, stroke, acute myocardial infarction, and cardiac arrest). We compared the demographics and comorbidities in both cohorts and extracted the odds ratio (multivariate analysis) of MACCE and associated in-hospital mortality. Results: Out of 1,141,120 geriatric obstructive sleep apnea G-OSA admissions, 9.9% (113,295) had MACCE. Males, Asians, or the Pacific Islander/Native American race, and patients from the lowest income quartile revealed a higher MACCE rate. Significant clinical predictors of MACCE in elderly OSA patients on multivariable regression analysis in decreasing odds were pulmonary circulation disease (OR 1.47, 95% CI 1.31–1.66), coagulopathy (OR 1.43, 95% CI 1.35–1.50), peripheral vascular disease (OR 1.34, 95% CI 1.28–1.40), prior sudden cardiac arrest (OR 1.34, 95% CI 1.11–1.62), prior myocardial infarction (OR 1.27, 95% CI 1.22–1.33), fluid and electrolyte imbalances (OR 1.25, 95% CI 1.20–1.29), male sex (OR 1.22, 95% CI-1.18–1.26), hyperlipidemia (OR 1.20, 95% CI 1.16–1.24), low household income (OR 1.19, CI 1.13–1.26), renal failure (OR 1.15, 95% CI 1.12–1.19), diabetes (OR 1.14, 95% CI 1.10–1.17), metastatic cancer (OR 1.14, 95% CI 1.03–1.25), and prior stroke or TIA (OR 1.12, 95% CI 1.07–1.17) (All *p* value < 0.05). Conclusions: This study emphasizes the significant association between obstructive sleep apnea (OSA) and major cardiac and cerebrovascular events (MACCE) in the geriatric population. Among the elderly OSA patients, a substantial 9.9% were found to have MACCE, with specific demographics like males, Asian or Pacific Islander/Native American individuals, and those from the lowest income quartile being particularly vulnerable. The study sheds light on several significant clinical predictors, with pulmonary circulation disease, coagulopathy, and peripheral vascular disease topping the list. The highlighted predictors provide valuable insights for clinicians, allowing for better risk stratification and targeted interventions in this vulnerable patient cohort. Further research is essential to validate these findings and inform how tailored therapeutic approaches for geriatric OSA patients can mitigate MACCE risk. Clinical Implications: Elderly individuals with a high risk for MACCE should undergo routine OSA screening using tools like the sensitive STOP-BANG Questionnaire. Implementing CPAP treatment can enhance cardiovascular outcomes in these patients.

## 1. Introduction

Obstructive sleep apnea (OSA) is a prevalent condition among the elderly, with a reported prevalence rate of 35.9% [1]. As individuals age, the incidence of major adverse cardiovascular and cerebrovascular events (MACCE) escalates significantly, with approximately 62% stroke hospitalizations and 60–65% myoardial infarction occurring in those aged 65 and above [2]. Obstructive sleep apnea (OSA) is a sleep disorder that affects elderly and often obese individuals [3]. This condition is defined by repeated episodes of the airway completely collapsing while sleeping, leading to low oxygen levels and disrupted sleep patterns [4]. OSA has been associated with an increased likelihood of events such as myocardial infarction, strokes, and sudden cardiac arrests [5]. According to the latest European Guidelines on cardiovascular disease, obstructive sleep apnea (OSA) has emerged as a significant factor influencing the risk of major adverse cardiovascular and cerebrovascular events (MACCE) [6]. OSA has been associated with different forms of cardiovascular disease, including hypertension, stroke, and coronary artery disease. However, we still lack information regarding the prevalence and impact of major adverse cardiac and cerebrovascular events (MACCE) in elderly patients with OSA (referred to as G OSA). Previous studies have suggested that conditions like hypertension, diabetes, and obesity can contribute to the development and progression of OSA in the population. Moreover, these underlying health cardiovascular disease risk factors are known to heighten the risk of events for individuals experiencing OSA. Consequently, it is crucial for us to determine the prevalence and factors predicting MACCE in G OSA patients in order to establish management strategies and minimize negative outcomes.

In this study, our goal was to examine the prevalence and predictors of MACCE among hospitalized G OSA patients using data from the National Inpatient Sample (NIS) database. We hypothesized that G OSA patients who have existing cardiovascular disease risk factors or comorbidities would face a risk of experiencing MACCE compared to those without these risk factors. This population-based study will provide insights into the impact of MACCE in patients with G OSA and identify key factors that can assist in assessing risk levels and devising effective management approaches. This can lead to better outcomes and decreased healthcare expenses for this vulnerable population.

## 2. Methods

### 2.1. Study Design and Setting

This study utilized data from the National Inpatient Sample (NIS) database, which is sponsored by the Agency for Healthcare Research and Quality (AHRQ). The NIS contains discharge records for over 7 million hospitalizations across 45 states in the United States gathered from more than 1000 federal institutions [7]. The focus of this research was on hospitalized patients who were diagnosed with obstructive sleep apnea (OSA) and were aged 65 years or older.

### 2.2. Study Population and Data Collection

The participants were divided into two groups based on whether they experienced cerebrovascular events (referred to as MACCE) or not. MACCE includes events such as death from any cause, stroke, myocardial infarction, and cardiac arrest. We compared the demographics, comorbidities, and clinical characteristics of these two groups. Through multivariable logistic regression analysis, we calculated the odds ratio (OR) of MACCE and its connection to hospital mortality.

### 2.3. Statistical Analysis

Descriptive statistics were used to summarize the demographics and comorbidities of the study population. Differences in baseline characteristics between the two cohorts were assessed using chi-squared tests for categorical variables and Student’s *t*-tests for continuous variables. A multivariate logistic regression analysis was performed to determine the predictors of MACCE and associated in-hospital mortality. The results were reported as adjusted odds ratios (aORs) with 95% confidence intervals (CIs). The NIS is a publicly available database that does not contain patient identifiers; therefore, this study was exempt from Institutional Review Board (IRB) approval. The IBM SPSS Statistics version 25.0 (IBM Corp., Armonk, NY, USA) was used to perform all statistical tests using complex sample modules and strata/cluster designs.

## 3. Results

A total of 1,141,120 geriatric hospitalizations with OSA were identified in the National Inpatient Sample, of which 9.9% (113,295) had major adverse cardiac and cerebrovascular events (MACCE) during their hospitalization. The MACCE cohort had a higher proportion of males (63.6% vs. 57%) and was relatively older (median age 74 vs. 73 years) compared to the non-MACCE cohort.

Patients from the lowermost income quartile had a higher MACCE rate compared to those from the highest income quartile (OR 1.19, 95% CI 1.13–1.26). Similarly, Asian or Pacific Islander/Native American race (OR 1.10, 95% CI 1.05–1.15) and male sex (OR 1.22, 95% CI 1.18–1.26) were significantly associated with increased odds of MACCE.

The prevalence of several comorbidities, including congestive heart failure (27% vs. 26.1%), hyperlipidemia (68% vs. 62.9%), diabetes mellitus (55.7% vs. 49.7%), peripheral vascular disease (12.9% vs. 9.3%), prior myocardial infarction (15.6% vs. 11.1%), prior stroke or TIA (12.3% vs. 10.6%), coagulopathy (10.7% vs. 7%), renal failure (39.4% vs. 32.1%), pulmonary circulation disease (1.7% vs. 1.1%), deficiency anemias (25.2% vs. 23.4%), fluid/electrolyte imbalance (39.4% vs. 30.8%), and alcohol abuse (2.1% vs. 1.8%), was significantly higher in the MACCE cohort than in the non-MACCE cohort (all *p* < 0.05) (Table 1).

In the context of multivariable regression analysis, several factors were found to be significant predictors of major adverse cardiovascular and cerebrovascular events (MACCE). These factors include pulmonary circulation disease (OR 1.47, 95% confidence interval CI 1.31–1.66), coagulopathy (OR 1.43, 95% CI 1.35–1.50), peripheral vascular disease (OR 1.34, 95% CI 1.28–1.40), prior sudden cardiac arrest (OR 1.34, 95% CI 1.11–1.62), prior myocardial infarction (OR 1.27, 95% CI 1.22–1.33), fluid and electrolyte imbalances (OR 1.25, 95% CI 1.20–1.29), male sex (OR 1.22, 95% CI 1.18–1.26), hyperlipidemia (OR 1.20, 95% CI 1.16–1.24), low household income (OR 1.19, CI 1.13–1.26), renal failure (OR 1.15, 95% CI 1.12–1.19), diabetes (OR 1.14, 95% CI 1.10–1.17), metastatic cancer (OR 1.14, 95% CI 1.03–1.25), and prior stroke or TIA (OR 1.12, 95% CI 1.07–1.17) all *p* < 0.05 (Table 2).

## 4. Discussion

The primary aim of this research was to examine the frequency and predictability of major adverse cardiac and cerebrovascular incidents (MACCE) in elderly patients suffering from obstructive sleep apnea (G-OSA). Aging, in and of itself, is inherently associated with an elevated risk for major adverse cardiac and cerebrovascular events (MACCE) [8]. As individuals age, physiological changes combined with the accumulation of various risk factors over time predispose them to a range of cardiovascular complications. Obstructive sleep apnea (OSA), which sees increased prevalence with age, further compounds this risk.

The increased prevalence of OSA in the elderly is attributed to several age-related physiological changes. Specifically, the narrowing of the upper airway lumen is especially pronounced in males. Structural alterations, such as the elongation of the pharyngeal airway across both genders and the descent of the hyoid bone—more evident in those with elongated facial structures—contribute to increased pharyngeal resistance and a predisposition toward airway constriction, even in the absence of other health complications [9,10].

Moreover, the elderly demonstrate an increased frequency of sleep arousals, fostering respiratory instability [11] and a diminished genioglossus muscle response to negative pressure and hypoxia. A noteworthy observation is the increased deposition of parapharyngeal fat, independent of overall body fat, which could further accentuate the OSA risk [10].

Delving into pathophysiology, obstructive sleep apnea (OSA) has emerged as a significant cardiovascular risk factor, primarily due to its characteristic episodes of upper-airway obstructions leading to intermittent oxygen deprivation during sleep [12]. The subsequent breathing irregularities and hypoxia stimulate sympathetic responses and oxidative stress, activating pro-inflammatory and pro-coagulant pathways [13]. Such physiological changes predispose individuals to endothelial dysfunction and adverse cardiovascular (CV) outcomes [14,15].

Several studies indicate a heightened prevalence of OSA in patients presenting with acute coronary syndrome (ACS), where it serves as an unfavorable prognostic factor [16,17,18,19]. In a notable multicenter study involving 1311 patients subjected to percutaneous coronary intervention, an OSA diagnosis was found to be an independent predictor of adverse CV events over a 2-year follow-up period [20]. Further research has bolstered this association, highlighting the positive impact of treating OSA on cardiovascular outcomes [21]. Additionally, higher apnea–hypopnea index (AHI) values correlated with an increased risk of ischemic stroke [22], while post-stroke CPAP therapy demonstrated reduced vascular events in OSA-compliant patients [23]. Despite these findings, the precise influence of OSA on the mortality rates of the elderly continues to be a subject warranting comprehensive exploration.

OSA in the elderly triggers heightened sympathetic activity, vascular oxidative stress, and inflammatory cascades due to recurrent hypoxemia [13]. Early studies linking OSA to cardiovascular conditions faced scrutiny, but subsequent research reinforced the relationship, highlighting improvements in OSA treatment [13,24]. For instance, a significant increase in ischemic stroke risk was noted with higher apnea–hypopnea index (AHI) values [22], and CPAP therapy post-stroke showed reduced vascular events in compliant OSA patients [23]. Our research primarily focuses on predictors of MACCE within this demographic.

Although numerous studies have acknowledged OSA as a risk factor for MACCE, our study is the first to specifically determine the predictors of MACCE in the elderly population with OSA. This understanding allows us to address modifiable predictors, customize management, and design interventions tailored for specific target populations. The study’s findings indicate that 10% of G-OSA admissions experienced MACCEs, highlighting a risk for these events among this population. Furthermore, the study identified comorbid factors that can help predict the occurrence of MACCE in G-OSA patients. In the following discussion, we will provide an analysis of the study’s results and discuss their implications.

One noteworthy discovery from the study was that being male increases the likelihood of MACCE in G-OSA patients. This aligns with research indicating an increased incidence of OSA among males compared to females [25]. Obstructive sleep apnea (OSA) affects men frequently, with a male-to-female ratio of about 2:1 in the general population and among older individuals [25]. There are potential factors that contribute to this discrepancy, including hormonal influences, differences in body fat distribution, and variations in pharyngeal anatomy [25]. Similar to the prospective cohort studies by Punjabi et al. and Marin et al., our study showed male gender as a significant independent risk factor for MACCE among G-OSA patients [26,27]. Consequently, healthcare professionals should take a patient’s gender into account when assessing MACCE risk in G-OSA cases and developing management strategies. Another interesting finding from our research is that G-OSA patients with low household incomes are more likely to develop MACCE. The CPAP is the primary treatment for obstructive sleep apnea [28]. The use of the CPAP on a regular basis decreases the apnea–hypoxia index and daytime sleepiness, improves sleep quality, and reduces cardiovascular complications [10]. According to Simon Tuvol et al., low income is an independent predictor of poor CPAP acceptance, which leads to complications associated with OSA [28]. These findings highlight the need for policy implications to reinforce targeted CPAP support programs for low-income patients in order to increase CPAP acceptability and adherence, thereby minimizing OSA-related complications, including poor cardiovascular outcomes.

The study has also identified the prevalence of certain comorbidities that serve as predictive indicators for the incidence of MACCE (major adverse cardiovascular and cerebrovascular events) in patients diagnosed with G-OSA (obstructive sleep apnea). Notably, the prevalence of OSA is also higher among patients who concurrently suffer from comorbidities such as hypertension, coronary artery disease, atrial fibrillation, and stroke. Despite this heightened prevalence, it is noteworthy that a substantial proportion of these individuals remain underdiagnosed for OSA, rendering them vulnerable and at an elevated risk of experiencing MACCE [29]. Moreover, the presence of these comorbidities is associated with variations in symptomatology, polysomnography parameters, and cardiovascular risk profiles [30]. Furthermore, these comorbidities are recognized as independent risk factors for the development of MACCE. A distinguishing characteristic of OSA is the absence of a normal nocturnal drop in blood pressure [31]. In accordance with the findings of Wang et al., the preexistence of severe hypertension can significantly amplify the risk of MACCE in patients with OSA, particularly following an acute coronary syndrome, which can be attributed to the shared pathophysiological mechanisms of endothelial dysfunction and atherosclerotic plaque accumulation in both the conditions [32]. It is important to notice that a higher burden of comorbidities substantially worsens the prognosis for patients with OSA. Therefore, the timely and prompt utilization of continuous positive airway pressure (CPAP) therapy emerges as a crucial intervention strategy to mitigate the incidence of MACCE in this high-risk cohort [33].

Interestingly, our study revealed that in G-OSA patients, obesity is associated with a decreased risk of MACCE compared to non-obese patients. This paradoxical finding contradicts the widely accepted notion that obesity is a major risk factor for various health conditions, including cardiovascular disease [34]. The potential hypothesis behind the obesity paradox might be due to higher metabolic reserve in patients with a higher BMI, which aids in coping with acute myocardial infarction stress, and the protective effect of adipose tissue through the release of anti-inflammatory cytokines [35]. The obesity paradox, despite its existence in previous studies [35,36], is not widely accepted. Patients with OSA, particularly the elderly, should be checked on a regular basis by their providers, with an emphasis on addressing obesity, as obesity is a significant risk factor for chronic diseases such as cardiovascular events.

This study holds significant clinical implications, particularly in identifying geriatric patients with obstructive sleep apnea (G-OSA) who are at heightened risk of major adverse cardiac and cerebrovascular events (MACCE). Targeted screening and monitoring are imperative for individuals identified as high-risk, including males, individuals with low household incomes, and those with comorbidities. Although the general population does not undergo routine OSA screening, the elderly, given their elevated risk profile, present a compelling case for such evaluations. By emphasizing early detection within this demographic, we may reduce their risk of experiencing MACCE. A myriad of screening tools exists for OSA, such as the STOP-BANG Questionnaire, the Sleep Apnea Clinical Score, the Berlin questionnaire, the NoSAS score, and the multivariable apnea prediction instrument. Among these, the STOP-BANG Questionnaire has been recognized for its superior sensitivity [37]. Also, current research has illuminated several prognostic parameters of OSA concerning cardiovascular outcomes. Specifically, determinants such as an apnea–hypopnea index (AHI) of ≥15, a minimum arterial oxygen saturation (MinSaO_2_) of ≤85%, Epworth Sleepiness Scale scores of ≥11, and the onset of excessive daytime sleepiness have emerged as independent MACCE risk factors over a 3–5-year follow-up [38]. Guidelines also recommend that these intermediate outcomes can be ameliorated via OSA treatment with a CPAP [39]. Additionally, utilizing home sleep apnea testing (HSAT) as a cost-effective alternative to in-laboratory polysomnography (iPSG) can improve the accessibility of OSA diagnosis [40].

Ensuring accessible, continuous positive airway pressure (CPAP) therapy for individuals with low income is essential, requiring effective healthcare policies. Additionally, the efficient management of comorbidities is crucial, given their independent association with MACCE. A multidisciplinary approach involving sleep therapists and pulmonologists is advocated for comprehensive patient care. Timely diagnosis of OSA in the elderly necessitates resource allocation to reduce wait times for sleep studies and efficient management with CPAP titration. Educating individuals with OSA on managing comorbidities and adopting preventive measures, such as avoiding sedating medications and adopting proper sleep positions, is paramount.

One of the strengths of this study is its utilization of a sample of G-OSA patients that represents the national population. The findings from this study can be applied to a group of G-OSA patients, and the use of regression analysis strengthens the credibility of the results. However, it is important to acknowledge the limitations of our study. Firstly, there might be coding errors in the NIS database, which could lead to misclassifying OSA and MACCE. Variability in documentation and coding of comorbidities and outcomes across different healthcare facilities could influence the results. Secondly, our study only focuses on data from hospitalized patients. The database may not account for patients with undiagnosed or milder forms of OSA who do not require hospitalization, thereby limiting the generalizability of our findings. Thirdly, our research is limited to individuals aged 65 years and older; therefore, generalizing these findings to patients may not be appropriate. Another limitation of our study is that several other risk factors for OSA, such as nasal obstruction, polyps, chronic nasal inflammation, deviated nasal septum, tonsil hypertrophy, micro/retrognathia, macroglossia, neuromuscular disorders, acromegaly, hypothyroidism, and Cushing syndrome were not taken into consideration and could be potential confounders for MACCE risk. There has been an emerging association between the treatment of OSA and the improvement of lower urinary tract symptoms in patients with urological conditions. However, our study is limited in analyzing data on urological conditions. Finally, the diagnostic methods used for OSA in this retrospective analysis were not stratified by severity. As the severity of OSA can vary significantly, its impact on MACCE risk also differs accordingly. Future research could benefit from incorporating severity-specific analyses to better understand the relationship between OSA severity and MACCE risk in the geriatric population.

## 5. Conclusions

In conclusion, our study identified a high prevalence of MACCE in G-OSA patients and several predictors of MACCE, including male sex, lower household income, and various comorbidities. These predictors highlight the significance of comprehensive risk stratification within this vulnerable population, allowing for targeted care based on comorbidities in individuals requiring special attention [33]. The presence of comorbid conditions independently contributing to MACCE risk highlights the need for meticulous comorbidity management as an integral part of treatment. Our findings emphasize the importance of healthcare providers adopting a multidisciplinary and patient-centered approach involving targeted screening, early diagnosis, personalized management, patient education, and interdisciplinary collaboration. By addressing these risk factors and embracing a holistic perspective, we can enhance the quality of care for G-OSA patients, reduce MACCE incidence, and optimize healthcare resource allocation, ultimately improving patient outcomes and healthcare efficiency.

## Figures and Tables

**Table 1 medsci-11-00069-t001:** Baseline characteristics for major adverse cardiovascular and cerebrovascular events.

Variable		MACCE All-Cause Mortality, AMI, Cardiac Arrest and Stroke		Overall Geriatric OSA Admissions
		YES	NO	
		N = 113,295	N = 102,785	N = 1,141,120
Age (years)		73 (69–79)	74 (69–80)	73 (69–79)
		Yes	No	
Sex	Male	63.6%	57.0%	57.7%
	Female	36.4%	43.0%	42.3%
Race (Ref: White)	White	82.1%	82.5%	82.5%
	Black	9.6%	9.7%	9.7%
	Hispanic	4.9%	4.9%	4.9%
	Asian Pacific	1.5%	1.1%	1.1%
	Native Americans	0.4%	0.4%	0.4%
	Other Ethnicities	1.5%	1.4%	1.4%
Bed Size (Ref: Small)	Small	17.6%	21.3%	21.0%
	Medium	28.9%	28.5%	28.6%
	Large	53.5%	50.1%	50.5%
Location/Teaching status of hospital (Ref: Rural)	Rural	7.3%	8.6%	8.5%
	Urban non-teaching	18.6%	20.3%	20.2%
	Urban teaching	74.0%	71.1%	71.4%
Region of hospital (Ref: Northeast)	Northeast	14.6%	15.9%	15.8%
	Midwest	31.2%	30.6%	30.7%
	South	35.3%	35.6%	35.6%
	West	18.9%	17.9%	18.0%
Primary Expected Payer (Ref: Medicare)	Medicare	88.3%	89.5%	89.4%
	Medicaid	0.6%	0.6%	0.6%
	Private including HMO	7.9%	8.4%	8.4%
	Self-pay	0.3%	0.3%	0.3%
	No charges	0.0%	0.0%	0.0%
	Others	1.6%	2.4%	1.7%
Hypertension		63.7%	68.8%	68.3%
Diabetes Mellitus (with and without chronic complications)		55.7%	49.7%	50.3%
Hyperlipidemia		68.0%	62.9%	63.4%
Smoking		40.3%	42.4%	42.2%
Peripheral Vascular Disease		12.9%	9.3%	9.6%
Obesity		39.6%	41.8%	41.5%
Renal Failure		39.4%	32.1%	32.8%
Prior MI		15.6%	11.1%	11.6%
Prior PCI		1.4%	1.0%	1.0%
Prior CABG		13.7%	10.3%	10.7%
Prior TIA or stroke		12.3%	10.6%	10.7%
Prior SCA		0.6%	0.4%	0.4%
Prior history of VTE		6.5%	8.8%	8.5%
Personal history of cancer		14.1%	16.7%	16.4%
Congestive heart failure		27.0%	26.1%	26.2%
Valvular heart disease		9.1%	8.0%	8.1%
Pulmonary circulation disease		1.7%	1.1%	1.2%
Chronic pulmonary disease		39.2%	39.8%	39.8%
Liver disease		3.9%	4.3%	4.2%
Metastatic cancer		2.4%	2.1%	2.1%
Solid tumor without metastasis		2.8%	2.9%	2.9%
Rheumatoid arthritis		3.9%	5.2%	5.1%
Coagulopathy		10.7%	7.0%	7.4%
Fluid and electrolyte disorders		30.8%	39.4%	31.7%
Deficiency Anemias		25.2%	23.4%	23.5%
Alcohol Abuse		2.1%	1.8%	1.9%
Drug abuse		1.1%	1.3%	1.3%
Other neurological disorders		10.5%	12.2%	12.0%
Depression		15.9%	19.0%	18.7%
Age (years)				

PCI—percutaneous coronary intervention, CABG—coronary artery bypass surgery, TIA—transient ischemic attack, SCA—sudden cardiac arrest, VTE—venous thromboembolism. Other ethnicities—other races or multiple races.

**Table 2 medsci-11-00069-t002:** Predictors of major adverse cardiovascular and cerebrovascular events (MACCE).

Variable		Odds Ratio	95 Confidence		*p* Value
			Lower	Upper	
Sex	Male vs. Female	1.22	1.18	1.26	<0.001
Race (Ref: White)	Black	0.89	0.84	0.95	<0.001
	Hispanic	0.90	0.83	0.96	
	Asian Pacific	1.08	0.94	1.24	
	Native Americans	1.04	0.78	1.37	
	Others	1.05	0.93	1.19	
Bed Size (Ref: Small)	Medium	1.17	1.09	1.25	<0.001
	Large	1.25	1.18	1.33	
Location/Teaching status of hospital (Ref: Rural)	Urban non-teaching	1.15	1.05	1.25	<0.001
	Urban teaching	1.34	1.23	1.46	
Region of hospital (Ref: Northeast)	Midwest	1.11	1.02	1.20	<0.035
	South	1.06	0.98	1.15	
	West	1.12	1.02	1.21	
Median household income national quartile (Ref: 76–100th)	0–25th percentile	1.19	1.13	1.26	<0.001
	26–50th percentile	1.15	1.09	1.21	
	51–75th percentile	1.05	1.00	1.10	
Elective vs. non-elective admission	Non-elective vs. Elective admission	4.12	3.82	4.42	<0.001
Primary Expected Payer (Ref: Medicare)	Medicaid	0.88	0.72	1.08	<0.001
	Private including HMO	1.17	1.11	1.24	
Hypertension		0.89	0.86	0.92	<0.001
Diabetes Mellitus (with and without chronic complications)		1.14	1.10	1.17	<0.001
Hyperlipidemia		1.20	1.16	1.24	<0.001
Smoking		0.86	0.84	0.89	<0.001
Peripheral Vascular Disease		1.34	1.28	1.40	<0.001
Obesity		0.92	0.90	0.95	<0.001
Renal Failure		1.15	1.12	1.19	<0.001
Prior MI		1.27	1.22	1.33	<0.001
Prior PCI		1.16	1.02	1.32	0.020
Prior CABG		1.09	1.04	1.14	<0.001
Prior TIA or stroke		1.12	1.07	1.17	<0.001
Prior SCA		1.34	1.11	1.62	0.003
Prior history of VTE		0.73	0.69	0.77	<0.001
Personal history of cancer		0.82	0.78	0.85	<0.001
Congestive heart failure		0.87	0.84	0.90	<0.001
Valvular heart disease		1.11	1.05	1.17	<0.001
Pulmonary circulation disease		1.47	1.31	1.66	<0.001
Chronic pulmonary disease		0.92	0.89	0.94	<0.001
Liver disease		0.80	0.74	0.86	<0.001
Acquired immune deficiency syndrome		0.18	0.04	0.74	0.018
Metastatic cancer		1.14	1.03	1.25	0.013
Solid tumor without metastasis		0.88	0.81	0.96	0.004
Rheumatoid arthritis		0.81	0.75	0.87	<0.001
Coagulopathy		1.43	1.35	1.50	<0.001
Fluid and electrolyte disorders		1.25	1.20	1.29	<0.001
Deficiency Anemias		0.89	0.86	0.92	<0.001
Alcohol Abuse		1.10	0.99	1.22	0.070
Drug abuse		0.87	0.76	1.00	0.048
Other neurological disorders		0.82	0.78	0.86	<0.001
Depression		0.86	0.82	0.89	<0.001
Age (years)		1.01	1.00	1.01	<0.001

PCI—percutaneous coronary intervention, CABG—coronary artery bypass surgery, TIA—transient ischemic attack, SCA—sudden cardiac arrest, VTE—venous thromboembolism.

## Data Availability

The data utilized in this research are available from the author upon request. The information is not available to the public because of privacy restrictions.

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
