# Peer review of "Major Adverse Cardiac and Cerebrovascular Events in Geriatric Patients with Obstructive Sleep Apnea: An Inpatient Sample Analysis"

_medsci, 2023, doi:10.3390/medsci11040069_

Round 1
Reviewer 1 Report
Comments and Suggestions for Authors
Authors should be congratulated for the topic. The manuscript is well-written and the methodology is robust. Moreover, the results achieved are clinically relevant and they could change the management of OSAS patients. The mortality from cardiac heart events represents a major concern of our society. OSAS patients must be screened to reduce this phenomenon.
However, OSAS represented a multifactorial condition that must be analyzed precisely. First of all, the OSAS produced a urinary functioning imbalance due to several causes (PMID= 37167825). Secondly, LUTS represented a cause of Cardiovascular mortality (PMID= 35441910). As a result, this bothering condition must be properly studied because of the influence it could add to the analysis performed. Are data available on the urinary functioning and urological comorbidities of these patients?
This aspect must be addressed properly in the main text.
Reviewer 2 Report
Comments and Suggestions for Authors
The authors' attempt to elucidate the relationship between obstructive sleep apnea (OSA) and major cardiac and cerebrovascular events (MACCE) in geriatric patients is commendable, especially given the paucity of data in this area. However, some elements warrant closer scrutiny.
While the National Inpatient Sample from 2018 is a robust dataset, relying on a single-year snapshot may not capture the evolving nature of the disease and associated events. A multi-year analysis might provide a more comprehensive picture. The abstract presents a laundry list of clinical predictors without elaborating on the strength or significance of each, which could be misleading. For instance, are all predictors equally significant, or do some carry more weight?
It's also concerning that the potential underlying mechanisms linking OSA and MACCE in this demographic remain unexplored in the abstract. Establishing such a direct relationship demands rigorous mechanistic explanations to strengthen the credibility of the findings.
Lastly, while the authors suggest routine screening for high-risk G-OSA patients, the abstract doesn't provide concrete recommendations on how this can be executed or what preventive measures can be adopted post-screening.
In summary, while the study offers preliminary insights into the association between OSA and MACCE in elderly patients, the abstract could benefit from a more nuanced presentation of findings and clearer actionable insights for clinical implementation.
Comments on the Quality of English LanguageMinor editing of English language is required.
Reviewer 3 Report
Comments and Suggestions for Authors
The present paper aimed to examine the prevalence and predictors of major cardiac and cerebrovascular events (MACCE) among hospitalized geriatric patients with obstructive sleep apnea (GOSA) using data from the National Inpatient Sample (NIS) database. The authors hypothesized that GOSA patients who have existing cardiovascular disease risk factors or comorbidities would face a risk of experiencing MACCE compared to those without these risk factors.
A few changes are needed, as follows:
Please explain every abbreviation before using it!
Introduction: Please mention that OSA is a clinical condition influencing CV risk according to the 2021 European Guidelines on cardiovascular disease in clinical practice (Visseren FLJ, et al.; ESC National Cardiac Societies; ESC Scientific Document Group. 2021 ESC Guidelines on cardiovascular disease prevention in clinical practice. Eur Heart J. 2021 Sep 7;42(34):3227-3337. doi: 10.1093/eurheartj/ehab484. Erratum in: Eur Heart J. 2022 Nov 7;43(42):4468.)
Results: You do not have to include all the OR and CI in the text. It is enough if you have them in the table. Just mention all statistically significant results.
Table 1. Column 2 is not clear…what does it represent?
Table 2: What do “Others” include?
Discussion: Please rephrase the first sentence!
Study limitations: Several other comorbidities, considered risk factors for OSA, were not considered, such as: nasal obstruction, polyps, chronic nasal inflammation, allergic rhinitis, deviated nasal septum, tonsils hypertrophy, micro/retrognathia, macroglossia, neuromuscular disorders, Acromegaly, Hypothyroidism and Cushing syndrome. Please mention it as a study limitation.
Please emphasize what is new in your study!
Conclusions: Lines 222-223: “Utilizing home sleep apnea testing (HSAT) as a cost-effective alternative to in-laboratory polysomnography (iPSG) can improve OSA diagnosis accessibility.” Please remove this statement from Conclusions and include it in Discussion.
Round 2
Reviewer 1 Report
Comments and Suggestions for Authors
Authors addressed properly their review